# Crystal structure of the full Swi2/Snf2 remodeler Mot1 in the resting state

Agata Butryn[1,2†], Stephan Woike[1,2], Savera J Shetty[3], David T Auble[3], Karl-Peter Hopfner[1,2,4*]

[1]Department of Biochemistry, Ludwig-Maximilians-Universität München, Munich, Germany; [2]Gene Center, Ludwig-Maximilians-Universität München, Munich, Germany; [3]Department of Biochemistry and Molecular Genetics, University of Virginia Health System, Charlottesville, United States; [4]Center for Integrated Protein Sciences Munich, Munich, Germany

**Abstract** Swi2/Snf2 ATPases remodel protein:DNA complexes in all of the fundamental chromosome-associated processes. The single-subunit remodeler Mot1 dissociates TATA box-binding protein (TBP):DNA complexes and provides a simple model for obtaining structural insights into the action of Swi2/Snf2 ATPases. Previously we reported how the N-terminal domain of Mot1 binds TBP, NC2 and DNA, but the location of the C-terminal ATPase domain remained unclear (*Butryn et al., 2015*). Here, we report the crystal structure of the near full-length Mot1 from *Chaetomium thermophilum.* Our data show that Mot1 adopts a ring like structure with a catalytically inactive resting state of the ATPase. Biochemical analysis suggests that TBP binding switches Mot1 into an ATP hydrolysis-competent conformation. Combined with our previous results, these data significantly improve the structural model for the complete Mot1:TBP:DNA complex and suggest a general mechanism for Mot1 action.
DOI: https://doi.org/10.7554/eLife.37774.001

*For correspondence:
hopfner@genzentrum.lmu.de

Present address: †Diamond Light Source Limited, Harwell Science and Innovation Campus, Didcot, United Kingdom

Competing interests: The authors declare that no competing interests exist.

## Introduction

Swi2/Snf2 ATPases are members of the NTP-dependent helicase/translocase superfamily 2 (SF2) and are well known as the principal ATP hydrolyzing 'engines' of chromatin remodelers that govern processes such as transcription, replication, and DNA repair (*Flaus et al., 2006*; *Narlikar et al., 2013*; *Hopfner et al., 2012*; *Becker and Workman, 2013*). It is generally assumed, that the Swi2/Snf2 ATPase motor translocates on the minor groove of double-stranded DNA and that this universal core activity generates the force for the large diversity of remodeling reactions catalyzed by Swi2/Snf2 proteins (*Saha et al., 2002*; *Whitehouse et al., 2003*; *Zofall et al., 2006*; *Dürr et al., 2005*). However, very little is known about how groove tracking activity is converted into the diverse chemo-mechanical remodeling reactions (*Hauk and Bowman, 2011*; *Narlikar et al., 2013*; *Blossey and Schiessel, 2018*). In the absence of substrates, remodelers have been observed in catalytically inactive resting states (*Hauk et al., 2010*; *Xia et al., 2016*; *Yan et al., 2016*), but it is unclear how universal auto-inhibited resting states are. Recent work provides some insight into how Swi2/Snf2 chromatin remodelers interact with and reconfigure nucleosomal substrates (*Liu et al., 2017*; *Farnung et al., 2017*; *Ayala et al., 2018*; *Eustermann et al., 2018*; *Sundaramoorthy et al., 2018*). However, the architecture and chemo-mechanical mechanisms of the diverse types of remodeling reactions are not well understood for the great majority of enzymes in this class.

The single subunit remodeler Mot1 (Modifier of transcription 1) is a Swi2/Snf2 enzyme that either activates or represses transcription in a context-dependent manner by dissociating TATA box-binding protein (TBP) and Negative Cofactor 2 (NC2) from promoter DNA (*Dasgupta et al., 2002*; *Zentner and Henikoff, 2013*). Mot1 is an essential Swi2/Snf2 enzyme in yeast and the Mot1-TBP-

NC2 regulatory axis is highly conserved in eukaryotes. The 140 – 210 kDa Mot1 protein has two functional domains. The N-terminal domain (Mot1$^{NTD}$) recognizes TBP while the C-terminal domain (Mot1$^{CTD}$) contains the catalytic Swi2/Snf2 ATPase that binds the DNA upstream from the TATA box (*Moyle-Heyrman et al., 2012*). The structure of the Mot1$^{NTD}$ has been determined by X-ray crystallography in complex with TBP (*Wollmann et al., 2011*) and in complex with TBP:NC2:DNA (*Butryn et al., 2015*). Low-resolution negative stain electron microscopy and chemical crosslinking coupled to mass spectrometry indicated the approximate location of the Mot1$^{CTD}$ near the opening of the Mot1$^{NTD}$ horseshoe (*Butryn et al., 2015*). However, the orientation of the Swi2/Snf2 domain and consequently the path of DNA remained elusive. As a result, the mechanism of Mot1-mediated dissociation of TBP complexes is still not well understood.

Here, we report the crystal structure of the near full-length Mot1 protein from *Chaetomium thermophilium*. Our structure reveals the location and orientation of the Swi2/Snf2 domain and, supported by mutagenesis studies, suggests a new type of resting state. Our data allow us also to derive a model for the Mot1 remodeler in complex with TBP and DNA.

## Results and discussion

### Architecture of *Ct*Mot1

We crystallized the near full-length Mot1 protein from *Chaetomium thermophilum*. The construct covers the entire Mot1$^{NTD}$ and Mot1$^{CTD}$ domains but lacks 50 amino acids from the C-terminus (*Figure 1A*). We determined the structure of this construct (residues 1–1836, Mot1$^{ΔC}$) harboring a point mutation in the Walker B motif (E1434Q) by Se-SAD to 3.2 Å (*Table 1*).

The *Ct*Mot1 enzyme is a ring-shaped protein (*Figure 1B*). The *Ct*Mot1$^{NTD}$ consists of 16 HEAT repeats (HR) with insertions at four sites and is similar to the much smaller *Encephalitozoon cuniculi* orthologue (*Ec*Mot1$^{NTD}$) with some notable differences. The helices forming the HEAT repeats are not extended in number but in length and the insertion elements into the HEAT repeats are longer. Thus, genome compaction in *E. cuniculi* did not alter the overall architecture of Mot1, which appears to be highly conserved in evolution, consistent with its critical function.

The structure reported here is the first to visualize the position and orientation of the Swi2/Snf2 ATPase domain in Mot1. The ATPase domain contains two characteristic lobes connected by a short hinge helix. Each lobe consists of a RecA-like subdomain (1A or 2A) that harbors the SF2-specific sequence motifs responsible for ATP and DNA binding as well as Swi2/Snf2-specific helical subdomains 1B, 2B, and 'brace' that emanate from 1A, 2A and the C-terminus of 2A, respectively. Lobe 1 of the *Ct*Mot1$^{CTD}$ contacts the C-terminus of *Ct*Mot1$^{NTD}$ via HR16, a small insertion within HR12, and a ~ 45 amino acids linker. This highly hydrophobic surface has a total area of 2500 Å$^2$. The tip of subdomain 1B interacts with HR1, thus lobe 1 effectively closes the ring structure of *Ct*Mot1. The architectural constraints imposed by a ring explain the conservation of the number of HEAT elements among Mot1 proteins. Lobe 2 binds the cleft between lobe 1 and HR1/2. The ~1900 Å$^2$ interface between lobe 2 and the remainder of *Ct*Mot1 is dominated by hydrogen bonds and salt bridges.

In some remodelers, the brace is directly followed by a 'bridge' element (*Hauk et al., 2010*), also referred to as NegC (*Clapier and Cairns, 2012*) or SnAC (*Sen et al., 2011*; *Xia et al., 2016*). While *Ec*Mot1 does not possess this element, in *Ct*Mot1 it is 64 amino acids long (residues 1822 – 1886). The bridge can act as a positive or negative auto-regulatory element via mechanisms that are not understood (*Wang et al., 2014*; *Xia et al., 2016*; *Yan et al., 2016*; *Clapier and Cairns, 2012*; *Carroll et al., 2014*; *Sen et al., 2011*). The bridge was almost entirely omitted from our crystallization construct and the only included residues (1822 –1836) together with the C-terminal expression tag are not visible in the electron density maps.

In summary, the structure reveals the architecture of the *Ct*Mot1 protein. It forms a ring-like structure in which the substrate-interacting HEAT repeat 'arch' binds both lobes of the ATPase domain from opposing sites.

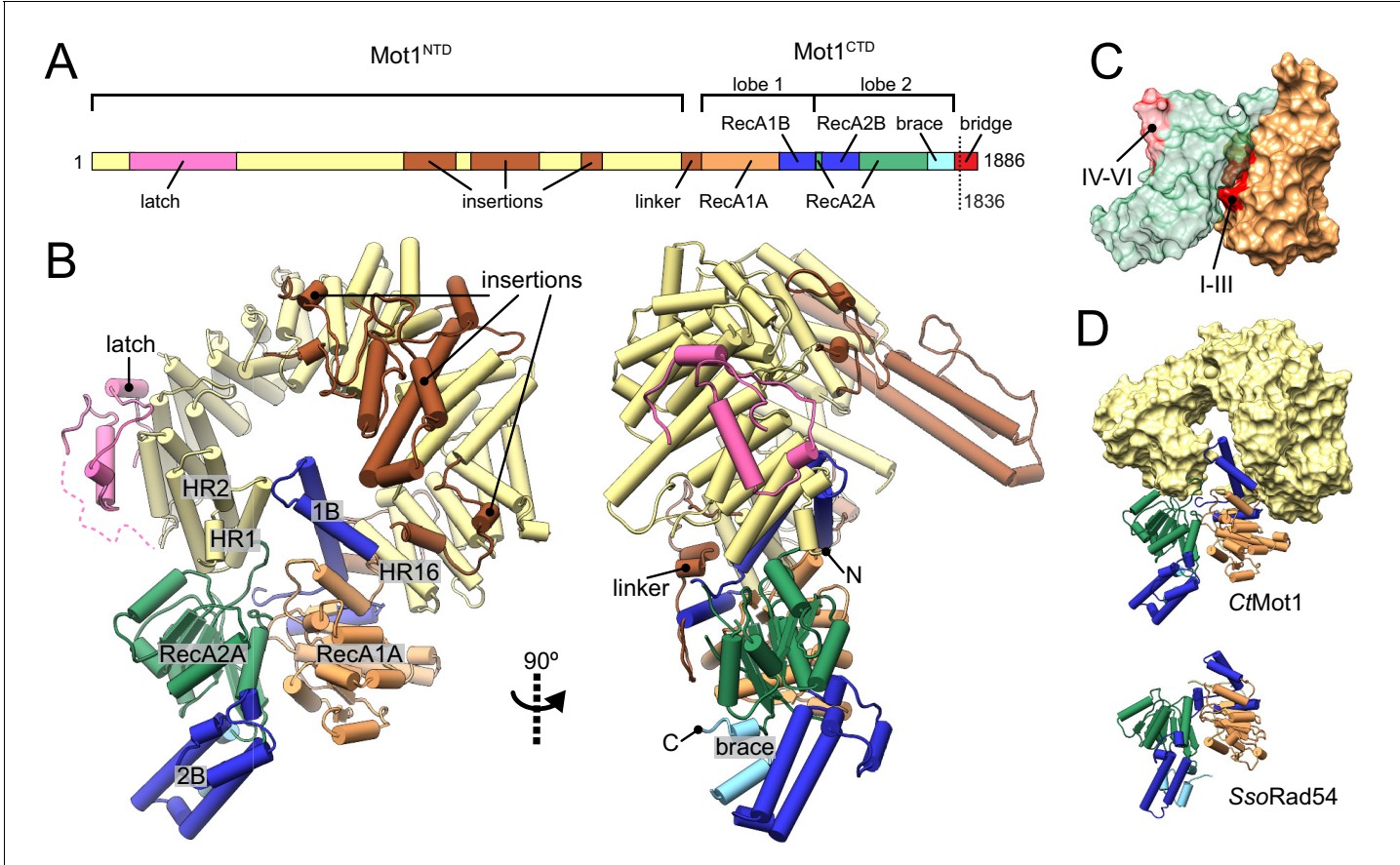

**Figure 1.** Structure of the *Chaetomium thermophilum* Mot1. (A) Domain organization of *Ct*Mot1. The HEAT repeats are in yellow. The latch is in pink, other insertions of *Ct*Mot1^NTD and the linker are in brown. RecA-like subdomains of *Ct*Mot1^CTD are in orange (1A) and green (2A). Swi2/Snf2-specific insertions 1B and 2B are in dark blue. Brace and bridge elements are in light blue and red, respectively. The boundary of the crystallization construct (residue 1836) is marked with the dotted line. (B) Cartoon representation of the structure. N- and C-termini are labelled N and C, respectively. HEAT repeats 1, 2, and 16 are labelled HR1, HR2, and HR16, respectively. Missing residues of the latch are represented by the dotted line. (C) Surface representation of *Ct*Mot1^CTD lobe 1 (orange) and 2 (green). Regions where helicase motifs are located on each lobe are colored in red. (D) Side-by-side comparison of *Ct*Mot1^CTD (top panel) and *Sso*Rad54 (*Dürr et al., 2005*) (bottom panel). *Ct*Mot1^NTD is represented as yellow surface. If not stated otherwise, all panels have color coding as in A.

DOI: https://doi.org/10.7554/eLife.37774.002

The following figure supplement is available for figure 1:

**Figure supplement 1.** Auto-inhibited conformations of Swi2/Snf2 domains.
DOI: https://doi.org/10.7554/eLife.37774.003

## Apo *Ct*Mot1 adopts an auto-inhibited resting state

In all species tested (*H. sapiens*, *S. cerevisiae*, *C. thermophilum*, *E. cuniculi*), Mot1's ATPase is robustly activated by TBP:DNA complexes, but very little if at all by DNA alone (*Auble et al., 1997*; *Adamkewicz et al., 2000*; *Wollmann et al., 2011*; *Chicca et al., 1998*). Interestingly, some Mot1 species are activated by TBP alone and do not require DNA, although a more robust activation is generally observed in the presence of both DNA and TBP. This suggests that the conformation of the Mot1^CTD is structurally coupled to TBP binding to the Mot1^NTD and that Mot1 alone is in an inactive state (*Adamkewicz et al., 2000*; *Moyle-Heyrman et al., 2012*). Indeed, comparison of *Ct*Mot1^CTD to other SF2 enzymes shows that lobe 2 is flipped ~180° from an 'active' conformation in which the ATPase and DNA-binding motifs would be properly aligned, that is lobe 1's motifs I-III are properly situated in the ATP-binding cleft, while lobe 2's motifs IV-VI are situated on the outside and are fully solvent-exposed (*Figure 1C*). As more Swi2/Snf2 domain structures have become available, it has become evident that many show an auto-inhibited conformation with misaligned lobes 1 and 2

**Table 1.** Data collection and refinement statistics for the *Ct*Mot1 structure.

| Data collection | |
|---|---|
| Space group | P2$_1$ |
| Unit cell | |
| a, b, c (Å) | 93.2, 96.9, 129.7 |
| α, β, γ (°) | 90.0, 97.6, 90.0 |
| Resolution (Å) | 48.7 (3.3–3.2)[*] |
| Total reflections | 239071 (10913) |
| Unique reflections | 36422 (1888) |
| $R_{meas}$ [%] | 14.4 (88.7) |
| $I/\sigma I$ | 11.8 (2.6) |
| $CC_{1/2}$ | 0.99 (0.79) |
| Completeness (%) | 97.1 (68.8) |
| Redundancy | 6.6 (5.8) |
| | |
| Refinement | |
| Resolution (Å) | 48.7 (3.3–3.2) |
| No. reflections | 36410 (2930) |
| $R_{work}$ | 0.19 (0.42) |
| R$_{free}$ | 0.24 (0.42) |
| No. atoms | 12390 |
| Protein | 12390 |
| Ligand/ion | 0 |
| Water | 0 |
| $B$ factors (Å$^2$) | |
| Protein | 75 |
| Ligand/ion | |
| Water | |
| R.M.S deviations | |
| Bond lengths (Å) | 0.002 |
| Bond angles (°) | 0.463 |
| Ramachandran plot | |
| Favored [%] | 97 |
| Allowed [%] | 3 |
| Outliers [%] | 0 |

[*] Values in parentheses are for highest-resolution shell.

DOI: https://doi.org/10.7554/eLife.37774.004

(*Figure 1—figure supplement 1*) (*Dürr et al., 2005*; *Hauk et al., 2010*; *Xia et al., 2016*; *Yan et al., 2016*). For example, the DNA binding site of the *Saccharomyces cerevisiae* Chd1 Swi2/Snf2 domain is directly occluded by the chromodomain, providing a means of specific activation of the enzyme by interaction with a nucleosomal substrate (*Hauk et al., 2010*; *Farnung et al., 2017*) (*Figure 1—figure supplement 1A*). Intriguingly, the 'resting' conformation of *Ct*Mot1$^{CTD}$ is very similar to the crystallographic conformation of the *Sulpholobus solfataricus* Rad54 Swi2/Snf2 domain (*Dürr et al., 2005*) (*Figure 1D* and *Figure 1—figure supplement 1B*). The functional relevance of the *Sso*Rad54 Swi2/Snf2 domain conformation remained unclear because the crystallized and functionally analyzed fragment of *Sso*Rad54 comprised only the isolated Swi2/Snf2 domain.

Although the precise orientation of lobe 2 might be additionally determined by the bridge element that is missing in the structure, our structural data suggest that the auto-inhibited resting state of *Ct*Mot1 is stabilized by the interactions between subdomain 2A and HR1/2. To test this, we mutated ion pairs (R4-D1720, R45-D1716) and a hydrophobic loop (L1658A/Y1659A) to destabilize the resting state (*Figure 2A and B*). Basal ATPase activity of the point mutants was greatly increased compared to wild-type *Ct*Mot1 (WT) and was not further stimulated by DNA and TBP (*Figure 2C* and *Figure 2—figure supplement 1A*). *Ct*Mot1$^{\Delta C}$ did not show increased basal ATPase rates and was not activated by TBP alone. However, its ATPase activity in the presence of DNA-containing complexes exceeded that of the WT enzyme. To find out whether this elevated ATPase activity of the mutants translates into productive disruption of the substrate complexes, we performed remodeling assays. Notably, despite an increase in the ATP hydrolysis rate, the ability of *Ct*Mot1$^{\Delta C}$ to dissociate TBP:DNA complexes was impaired (*Figure 2D*). Assays performed under less efficient dissociation conditions that allowed the TBP:DNA complexes to persist confirmed that all other tested mutants (L1658A/Y1659A, R4D and D1720R) indeed behaved as the WT (*Figure 2—figure supplement 1B and C*). This shows that the bridge element acts in response to TBP binding and, similarly to SnAC in Snf2 (*Xia et al., 2016*), ensures productive coupling of ATP hydrolysis to the remodeling reaction.

Taken together, our data show that Mot1 adopts a resting state with low catalytic activity by stabilizing lobe 2 of the Swi2/Snf2 domain in an inactive conformation relative to lobe 1. Mobilization of lobe 2 from its auto-inhibited state explains the activation of Mot1's ATPase by TBP and TBP:DNA complexes.

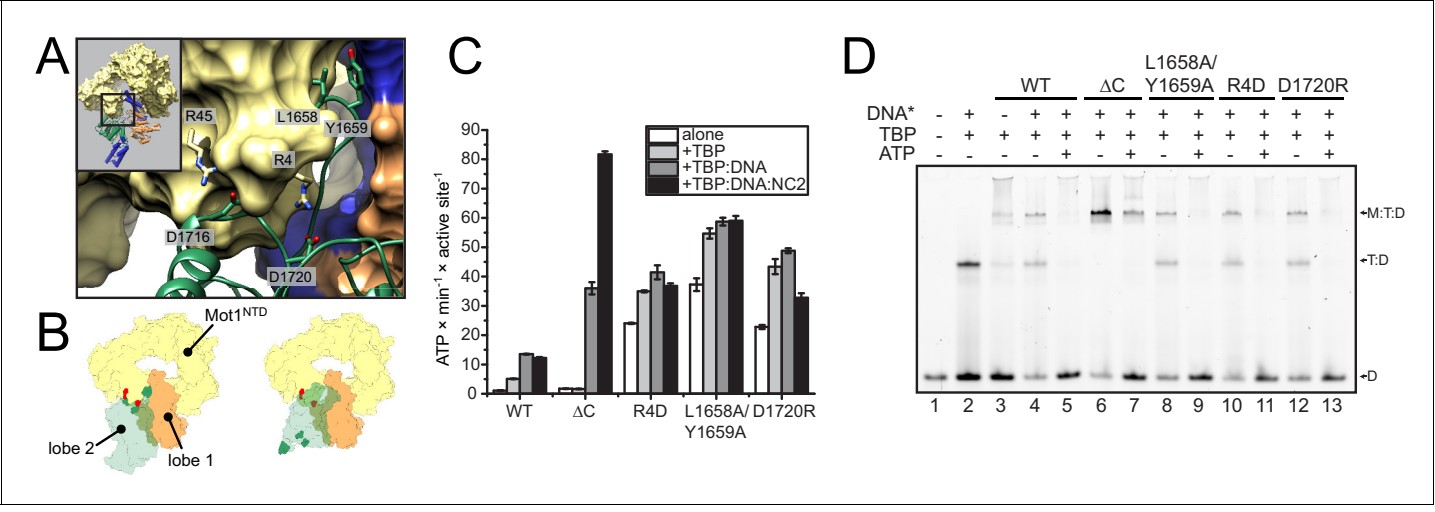

**Figure 2.** Analysis of *Ct*Mot1 mutants. (A) View at the interface between RecA2A (green cartoon), HR1/2 (yellow surface), and lobe 1 (orange/blue surface). Residues analyzed in this study are shown as sticks and labelled accordingly. (B) Cartoon model showing the positions of mutations in the Mot1$^{NTD}$ (red spheres on yellow surface) and in lobe 2 (green spheres on green surface). Left: orientation as in the *Ct*Mot1 crystal structure. Right: *Ct*Mot1 with ATPase modeled as in the *S. cerevisiae* Chd1:nucleosome complex, that is the ATP hydrolysis-competent conformation (*Farnung et al., 2017*). (C) ATPase activity of the mutants. Error bars represent standard deviations from three technical replicates. *Ct*Mot1$^{WT}$ is labelled as WT, *Ct*Mot1$^{\Delta C}$ as ΔC. (D) Electrophoretic mobility shift assay showing ATP-dependent dissociation of Mot1:TBP:DNA and TBP:DNA complexes. All *Ct*Mot1 constructs form ternary complexes with labelled DNA and TBP (M:T:D). In the presence of ATP and unlabeled competitor DNA (DNA*), wild-type *Ct*Mot1 (WT), L1658/Y1659, R4D, and D1720R mutants fully disrupt M:T:D and T:D complexes (lanes 5, 9, 11, and 13, respectively), whereas *Ct*Mot1$^{\Delta C}$ (ΔC) is less efficient (lane 7).

DOI: https://doi.org/10.7554/eLife.37774.005

The following source data and figure supplement are available for figure 2:

**Source data 1.** Raw data from the ATPase activity assay used for *Figure 2C* and *Figure 2—figure supplement 1A*.
DOI: https://doi.org/10.7554/eLife.37774.007

**Source data 2.** Raw data from quantification of electrophoretic mobility shift assay used for *Figure 2—figure supplement 1B*.
DOI: https://doi.org/10.7554/eLife.37774.008

**Figure supplement 1.** Analysis of *Ct*Mot1 mutants.
DOI: https://doi.org/10.7554/eLife.37774.006

## Model of the Mot1:TBP:NC2:DNA complex

The new structure of the near full-length *Ct*Mot1 protein together with prior structures enables us to provide a model for the DNA path in the Mot1-bound protein:DNA complex (*Figure 3*). The *Ec*Mot1NTD:TBP:NC2:DNA complex can be readily superimposed with *Ct*Mot1 through the conserved structure of the HEAT repeats. Likewise, superimposing *Sso*Rad54:DNA with *Ct*Mot1ΔC via lobe 1 visualizes how the *Ct*Mot1 ATPase could initially contact duplex DNA since in all Snf2/Swi2 protein:substrate structures contacts between nucleic acid and the RecA1 subdomain are preserved. In the resulting model, localization and orientation of the Swi2/Snf2 domain is in full agreement with prior EM and CX-MS analyses (*Butryn et al., 2015*) and with biochemical studies showing that Mot1 covers two helical turns upstream from the TATA box (*Darst et al., 2001*; *Sprouse et al., 2006*; *Moyle-Heyrman et al., 2012*). Notably, the superimposed DNA segment bound by the ATPase is an almost direct continuation of the promoter DNA fragment from the *Ec*Mot1NTD:TBP:NC2:DNA crystal structure. Assuming the generally proposed directionality of ATP dependent translocation of Swi2/Snf2 motor domains on dsDNA (*Zofall et al., 2006*; *Saha et al., 2002*; *Whitehouse et al., 2003*), the structure of *Ct*Mot1 and the specific orientation of lobe 1 now suggests that the Swi2/Snf2 motor translocates 'towards' the TATA box and TBP along the nucleic acid scaffold.

Our model of the Mot1:TBP:NC2:DNA complex suggests where the Swi2/Snf2 domain of Mot1 might engage with upstream DNA and provides new insight into how ATP hydrolysis-associated events are coupled to dissociation of protein:DNA substrates. Since processive ATP-dependent translocase activity has not been observed in biochemical studies, Mot1 could exploit short-range tracking toward TBP. Given the immediate vicinity of the Swi2/Snf2 domain to TBP, very few

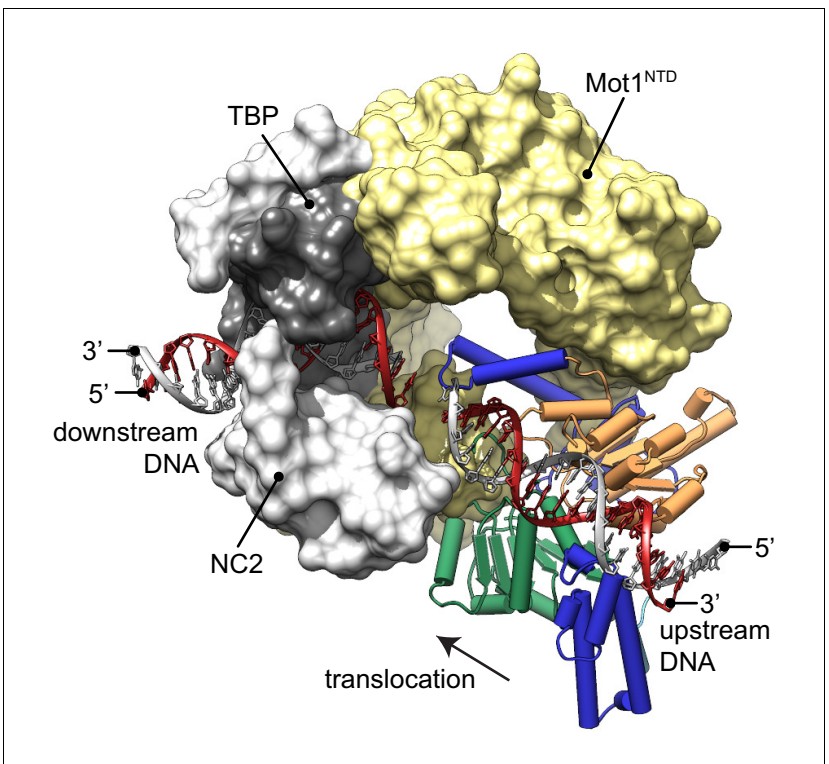

**Figure 3.** Model of the Mot1:TBP:NC2:DNA complex. *Ct*Mot1ΔC was superimposed onto *Ec*Mot1NTD:TBP:NC2:DNA via the HEAT repeats (yellow surface). The path of the upstream DNA was determined by superimposing *Sso*Rad54:DNA onto *Ct*Mot1ΔC via lobe 1. The TATA box strand from *Ec*Mot1NTD:TBP:NC2:DNA as well as the corresponding strand from *Sso*Rad54:DNA are marked in gray. The non-TATA box strands are in red. Substrate proteins TBP and NC2 are represented as dark and light gray surfaces, respectively. The black arrow represents the direction in which Swi2/Snf2 domain is proposed to translocate along the DNA scaffold. *Ct*Mot1CTD is color-coded as in *Figure 1*.
DOI: https://doi.org/10.7554/eLife.37774.009

translocation steps could lead to the displacement of TBP by steric collision (*Darst et al., 2001*; *Auble and Steggerda, 1999*; *Butryn et al., 2015*). In addition, Mot1 could simply displace TBP from DNA by overwinding or introducing other small distortions into upstream DNA (*Moyle-Heyrman et al., 2012*; *Butryn et al., 2015*). Similar effects have been observed for other transcription factors, for which not only binding but also dissociation rates depend on the structure of their recognition sites affected by the presence of other factors bound nearby (*Luo et al., 2014*; *Kim et al., 2013*). This allosteric effect can be accounted for by local changes to the major and minor groove width (*Kim et al., 2013*). Such a scenario is plausible since changes two helical turns upstream from the TBP binding site could have an immediate allosteric effect on severely bent and widened TATA box (*Tora and Timmers, 2010*).

Interestingly, while Mot1's ATPase orientation suggests that it 'pulls' DNA from TBP and overwinds DNA at the substrate, the reverse architecture is seen for the multisubunit INO80 remodeler: here the motor appears to pump DNA into the nucleosome and to underwind DNA at the substrate (*Eustermann et al., 2018*). Thus, our results suggest that Swi2/Snf2 proteins can use DNA translocation in different ways to disrupts protein:DNA interfaces.

# Materials and methods

## Key resources table

| Reagent type (species) or resource | Designation | Source or reference | Identifiers | Additional information |
|---|---|---|---|---|
| Gene (*Chaetomium thermophilum*) | CtMot1 | this paper | UniProtKB: G0S6C0 _CHATD | gene cloned from a cDNA library |
| Cell line (*Escherichia coli*) | Rosetta(DE3) | Novagen | Merck: 70954 | |
| Cell line (*Escherichia coli*) | B843(DE3) | Novagen | Merck: 69041 | |
| Recombinant DNA reagent | pETDuet-1 | Novagen | Merck: 71146 | used to express full-length CtMot1 (1–1886) and its point mutants |
| Recombinant DNA reagent | pET21b | Novagen | Merck: 69741 | used to express *Ct*Mot1 (1–1836) and *Ct*Mot1 (1–1836, E1434Q) |
| Chemical compound, drug | L(+)-Selenomethionine | Acros Organics | Acros Organics: 259960010 | 42 mg/mL final concentration |
| Chemical compound, drug | SelenoMethionine Medium Base plus Nutrient Mix | Molecular Dimensions | Molecular Dimensions: MD12-501 | |
| Chemical compound, drug | Adenosine 5′-triphosphate disodium salt hydrate (ATP) | Sigma-Aldrich | Sigma: A2383-10G | |
| Chemical compound, drug | β-Nicotinamide adenine dinucleotide reduced disodium salt hydrate (NADH) | Sigma-Aldrich | Sigma: 10107735001 | |
| Chemical compound, drug | Phospho(enol)pyruvic acid monopotassium salt (PEP) | PanReac AppliChem | AppliChem: A2271 | |
| Chemical compound, drug | Pyruvate kinase/lactic dehydrogenase enzymes from rabbit muscle | Sigma-Aldrich | Sigma: P0294 | |
| Software, algorithm | XDS | *Kabsch, 2010*, doi: 10.1107/S0907444909047374 | | |
| Software, algorithm | PHENIX | *Adams et al., 2010*, doi:10.1107/S0907444909052925 | | |

*Continued on next page*

*Continued*

| Reagent type (species) or resource | Designation | Source or reference | Identifiers | Additional information |
|---|---|---|---|---|
| Software, algorithm | *Coot* | *Emsley et al., 2010*, doi: 10.1107/S0907444910007493 | | |
| Software, algorithm | UCSF Chimera | *Pettersen et al., 2004* , doi: 10.1002/jcc.20084 | | http://www.rbvi.ucsf.edu/chimera/ |
| Software, algorithm | ImageJ 1.51 k | *Schneider et al., 2012*, doi: 10.1038/nmeth.2089 | | quantification of electrophoretic shift assay |
| Software, algorithm | OriginPro 2015 | OriginLab, Northampton, MA | | |
| Sequence-based reagent | 48 bp dsDNA | Biomers | | 5'–CAGTACGGCCG GGCGCCCGGCA TGGCGGCCTATAAAA GGGGGTGGAAT–3' |
| Sequence-based reagent | 48 bp 6-FAM labelled dsDNA | Biomers | | 5'–CAGTACGGCCGGGCGCCCG GCATGGCGGCCTATAAAA GGGGGTGGAAT–3' |
| Sequence-based reagent | 36 bp dsDNA | Biomers | | 5'–CGGCCGGGCGCCCGG CATGGCGGCCTAT AAAAGGGC–3' |

## Protein purification

The sequence of the full-length Mot1 (1 – 1886) was isolated from the *Chaetomium thermophilum* cDNA library and cloned into pETDuet-1 vector (Novagen, Germany) harboring N-terminal His$_6$ tag followed by TEV cleavage site. *Ct*Mot1$^{\Delta C}$(1 – 1836) was cloned into pET21 vector containing PreScission protease cleavage site and C-terminal His$_6$ tag. Both constructs were expressed in *Escherichia coli* Rosetta(DE3) cells (Novagen) and purified using Ni$^{2+}$-NTA agarose (QIAGEN, Germany). After proteolytic cleavage of the expression tags, the proteins were further purified using ion-exchange chromatography (HiTrap Q HP, GE Healthcare, Germany) and size exclusion chromatography (HiLoad 16/60 200 pg, GE Healthcare). Proteins were concentrated to ~15 mg/ml in 20 mM Tris pH 7.5, 50 mM NaCl and 15% glycerol and stored at −80°C. Selenomethionine labelling of *Ct*Mot1$^{\Delta C}$ was performed in *E. coli* B843 (Novagen) using SelenoMethionine Medium Base and Nutrient Mix (Molecular Dimensions, UK) supplemented with L(+)-Selenomethionine (Acros Organics, Germany) at 42 mg/L. Purification of selenium-derivatized protein was performed according to the same protocol as for the native protein.

## Crystallization and structure determination

Crystals of selenomethionine-derivatized *Ct*Mot1$^{\Delta C}$ were grown at 20°C by streak seeding in 0.1 M Tris pH 8.9, 0.2 M ammonium acetate and 13% (w/v) PEG 3350. Plate-like crystals with average dimensions of 700 × 150 × 30 μm appeared after three days and were cryocooled in liquid nitrogen using mother liquor supplemented with butanediol at 25% final concentration.

The data were collected at the European Synchrotron Radiation Facility in Grenoble, France at the peak of Se K-edge at 100K. Images were indexed, integrated, scaled, and merged in space group P2$_1$ to 3.2 Å using XDS package (*Kabsch, 2010*). The initial model was built manually to the experimental electron-density derived from SAD phasing using *PHENIX* AutoSol wizard (*Adams et al., 2010*). Alternating cycles of manual building using *Coot* (*Emsley et al., 2010*) and refinement with *PHENIX* yielded the final model (R$_{work}$/R$_{free}$ of 19.0/23.8%) covering 87% of all residues.

## ATPase assay

The assays were performed using an NADH-coupled assay as described (*Kiianitsa et al., 2003*). Reactions were performed using 2 mM phosphoenolpuryvate, 25 U/mL of pyruvate kinase/lactic dehydrogenase mix, 1 mM ATP, and 1 mM NADH at final concentrations. Test samples contained 100 nM dsDNA (5'–CAGTACGGCCGGGCGCCCGGCATGGCGGCCTATAAAAGGGGGTGGAAT–3' top strand), 100 nM TBP, 100 nM NC2 and 250 nM *Ct*Mot1.

## Electrophoretic mobility shift assays

Electrophoretic mobility shifts were essentially performed as described (*Darst et al., 2001*) with some modifications. In the assay shown in *Figure 2D*, fluorescently labelled dsDNA (40 nM, 5′–CAG TACGGCCGGGCGCCCGGCATGGCGGCCTATAAAAGGGGGTGGAAT–3′ top strand with 6-FAM label on the 5′ end of the reverse strand) was incubated with TBP (10 nM) and Mot1 (25 nM) for 10 min. Unlabeled dsDNA competitor (800 nM, 5′–CGGCCGGGCGCCCGGCATGGCGGCCTA TAAAAGGGC–3′ top strand) was then added directly followed by ATP addition (50 µM) for 10 min. Samples were loaded onto 6% polyacrylamide gels and run at 160 V and 4° for 60 min and imaged using Typhoon FLA 9000 imager. The assays shown in *Figure 2—figure supplement 1B* and quantified in *Figure 2—figure supplement 1C* were prepared analogously, but TBP was added at a concentration of 15 nM and ATP was added for 6 min before loading the reactions on the gel.

## Accession numbers

The coordinates and structure factors were deposited in the Protein Data Bank under accession code 6G7E.

## Acknowledgements

We thank the Max-Planck Crystallization Facility (Martinsried) for crystallization trials and the European Synchrotron Radiation Facility (Grenoble) and the Deutsches Elektronen-Synchrotron (PETRA III, Hamburg) for beamtime and excellent support. AB acknowledges support from the Integrated Analysis of Macromolecular Complexes and Hybrid Methods in Genome Biology (DFG GRK1721) training program.

## Additional information

### Funding

| Funder | Grant reference number | Author |
|---|---|---|
| National Institutes of Health | GM055763 | David T. Auble |
| European Commission | ERC Advanced Grant ATMMACHINE | Karl-Peter Hopfner |
| Deutsche Forschungsgemeinschaft | Gottfried Wilhelm Leibniz-Prize | Karl-Peter Hopfner |

The funders had no role in study design, data collection and interpretation, or the decision to submit the work for publication.

### Author contributions

Agata Butryn, Formal analysis, Investigation, Writing—original draft; Stephan Woike, Savera J Shetty, Investigation; David T Auble, Formal analysis, Writing—review and editing; Karl-Peter Hopfner, Conceptualization, Formal analysis, Supervision, Writing—original draft

### Author ORCIDs

Agata Butryn ⓘ http://orcid.org/0000-0002-5227-4770
Karl-Peter Hopfner ⓘ https://orcid.org/0000-0002-4528-8357

### Decision letter and Author response

Decision letter https://doi.org/10.7554/eLife.37774.015
Author response https://doi.org/10.7554/eLife.37774.016

## Additional files

### Supplementary files

• Transparent reporting form

DOI: https://doi.org/10.7554/eLife.37774.010

## Data availability

The coordinates and structure factors are deposited in the Protein Data Bank under accession code 6G7E. All data generated or analysed during this study are included in the manuscript and supporting files. Source data files have been provided for Figures 2 and Figure 2-figure supplement 1.

The following dataset was generated:

| Author(s) | Year | Dataset title | Dataset URL | Database, license, and accessibility information |
|---|---|---|---|---|
| Butryn A, Woike S, Shetty SJ, Auble DT, Hopfner K | 2018 | Crystal structure of the full Swi2/Snf2 remodeler Mot1 in the resting state | https://www.rcsb.org/structure/6G7E | Publicly available at the RCSB Protein Data Bank (accession no. 6G7E) |

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
