## [Decision Letter]

Thank you for submitting your article "Crystal structure of the full Swi2/Snf2 remodeler Mot1 in the resting state" for consideration by *eLife*. Your article has been reviewed by two peer reviewers, one of whom is a member of our Board of Reviewing Editors, and the evaluation has been overseen by John Kuriyan as the Senior Editor. The following individual involved in review of your submission has agreed to reveal his identity: Gregory D Bowman (Reviewer #2).

The reviewers have discussed the reviews with one another and the Reviewing Editor has drafted this decision to help you prepare a revised submission.

Summary:

Mot1 is an important factor involved in TBP turnover and recycling, containing an Swi2/Snf2-type ATPase that is believed to disrupt TBP:DNA complexes. Previous high-resolution structural information showed how the HEAT repeat portion of Mot1 engaged TBP:DNA, yet how the ATPase motor was organized was not well defined. Here the authors present a crystal structure of the Mot1 regulator that includes the ATPase domains. The ATPase motor is in an inactive organization, with the two halves rotated significantly away from the expected active configuration for Swi2/Snf2 ATPases. Interestingly, this inactive state very much resembles that of Rad54, suggesting it may represent a common resting state. The authors support the idea that the structure shows an inhibited state using mutants at the observed interface, which activate ATPase activity. By combining this new structural data with the previous Mot1:TBP:DNA:NC2 complex, the authors propose an organization of the holocomplex, where the ATPase motor is positioned to engage and translocate on DNA toward TBP. This is an important advance to previous structural knowledge about this enzyme, and helps deepen our understanding of how Mot1 accomplishes its task.

Essential revisions:

1) Clear interpretation of the in vivo data is confounded by the result that the mutants that do not support viability also show substantially lower expression levels than WT. The mutant that supports viability (D1720R) appears to be the only mutant with expression levels comparable to WT. At the same time, to demonstrate a mechanistic advance the in vivo data is not really needed. So the authors should either remove the in vivo data from the revised manuscript or repeat the experiments under conditions where the expression levels of the mutants can be better controlled.

2) The authors suggest that Mot1 might displace TBP by "counteracting TBP-induced bending and underwinding", which makes sense given their structure. Others have demonstrated how adjacent DNA-binding factors can stimulate dissociation of their neighbors, likely by changing groove widths (Kim et al., 2013; Luo et al., 2014). A brief discussion of these systems would be beneficial in relation to the Mot1:TBP system, as the change in DNA geometry that the authors propose would likely be sufficient for rapid TBP eviction.

3) As mentioned by the authors, auto-inhibited states are being observed in a variety of different chromatin remodeling ATPases and Mot1 is distinct in terms of its substrate. It will therefore be useful to expand a bit the discussion in the first paragraph of the subsection “Apo *Ct*Mot1 adopts an auto-inhibited resting state”, by making a supplementary figure, which compares the misalignment of lobes 1 and 2 in the resting state structures of Chd1, SWI2 and ISWI to Mot1 in addition to the comparison with Rad54.

4) The data in Figure 2C should be quantified and error bars shown.

5) The reviewers don't think the authors can state "Our model of the Mot1:TBP:NC2:DNA complex illustrates how the Swi2/Snf2 domain of Mot1 engages with upstream DNA," since there is no DNA in the structure. It would be more appropriate to say "…suggests where…" instead of "illustrates how".

[Editors' note: further revisions were requested prior to acceptance, as described below.]

Thank you for resubmitting your work entitled "Crystal structure of the full Swi2/Snf2 remodeler Mot1 in the resting state" for further consideration at *eLife*. Your revised article has been favorably evaluated by John Kuriyan as the Senior Editor, and the Reviewing Editor.

The majority of the reviewer comments have been satisfactorily addressed. Before making a final decision we require one clarification regarding the gel in the new Figure 2D. It appears that in this gel, the ATP-dependent effects seen in lanes 5, 9, 11 and 13 mainly disrupt the ternary complex (M:T:D) and not the TBP-DNA complex (T:D). This seems to be different than in Wollman et al., 2011 and in the previous submission of this work where the T:D complex is disrupted in the presence of ATP and WT Mot1. Please send us a brief paragraph clarifying the reasons for these apparent differences.

---

## [Author Response]

Essential revisions:1) Clear interpretation of the in vivo data is confounded by the result that the mutants that do not support viability also show substantially lower expression levels than WT. The mutant that supports viability (D1720R) appears to be the only mutant with expression levels comparable to WT. At the same time, to demonstrate a mechanistic advance the in vivo data is not really needed. So the authors should either remove the in vivo data from the revised manuscript or repeat the experiments under conditions where the expression levels of the mutants can be better controlled.

We understand this concern. There is a SUMO- and ubiquitin-mediated protein quality control system in yeast whose function was uncovered through genetic analysis of Mot1 point mutants (Wang et al., Genetics 172:1499-1509, 2006; Wang and Prelich, MCB 29:1694-1706, 2009). This system is responsible for reduced levels of many mutant Mot1 proteins in vivo, even in some cases in which the amino acid changes are relatively subtle. We have also explored Mot1 function in vivo using multi-copy plasmids and strong regulated promoters such as GAL1, which do boost protein levels and can compensate for reduced protein levels. However, these over-expression systems often cause dominant-negative effects on growth, even when the over-expressed protein is wild-type Mot1 (Darst et al., 2003; Auble et al., 1997). Based on this prior work, while it would be possible to investigate the in vivo function of the point mutants included in the first submission by increasing their expression, we feel that such studies are unlikely to provide much additional insight into Mot1 function. For these reasons we have opted to remove the in vivo data at the reviewer’s suggestion.

2) The authors suggest that Mot1 might displace TBP by "counteracting TBP-induced bending and underwinding", which makes sense given their structure. Others have demonstrated how adjacent DNA-binding factors can stimulate dissociation of their neighbors, likely by changing groove widths (Kim et al., 2013; Luo et al., 2014). A brief discussion of these systems would be beneficial in relation to the Mot1:TBP system, as the change in DNA geometry that the authors propose would likely be sufficient for rapid TBP eviction.

We thank the reviewers for this interesting point of discussion. We expanded this part of the Discussion (subsection “Model of the Mot1:TBP:NC2:DNA complex”, second paragraph). We indicate that the proposed “counteracting” effect of the Snf2/Swi2 motor domain can be accounted for by an allosteric effect transmitted via DNA, a phenomenon that has been described for other transcription factors (Kim at al., 2013; Luo et al., 2014). Local changes to the DNA could be easily transmitted from the upstream DNA, where ATPase is located, to the highly constrained TATA box just two helical turn downstream.

3) As mentioned by the authors, auto-inhibited states are being observed in a variety of different chromatin remodeling ATPases and Mot1 is distinct in terms of its substrate. It will therefore be useful to expand a bit the discussion in the first paragraph of the subsection “Apo CtMot1 adopts an auto-inhibited resting state”, by making a supplementary figure, which compares the misalignment of lobes 1 and 2 in the resting state structures of Chd1, SWI2 and ISWI to Mot1 in addition to the comparison with Rad54.

As the reviewer suggested, we include a supplementary figure (Figure 1—figure supplement 1) and also expanded the Discussion (subsection “Apo *Ct*Mot1 adopts an auto-inhibited resting state”, first paragraph). The new figure (panel A) shows the comparison between the auto-inhibited and activated (nucleosome-bound) conformation of a Swi2/Snf2 domain illustrated by the example of Chd1. Panel B shows other Swi2/Snf2 domain structures representing different auto-inhibited conformations (Snf2, ISWI, Rad54 and Mot1). We additionally updated the colors in Figure 1A to match the updated scheme used in Figure 1—figure supplement 1.

4) The data in Figure 2C should be quantified and error bars shown.

We repeated electrophoretic mobility shift assays that were presented in Figure 2C and Figure 2—figure supplement 1B in the original submission file. We improved our procedure for performing these assays and we used polyacrylamide gels rather than agarose, which enhanced band sharpness and the overall signal to noise ratio. New results are shown in reorganised Figure 2 (an exemplary gel, panel D) and Figure 2—figure supplement 1 (quantification from six technical replicates, panel B). We did not repeat the assays on TBP:DNA:NC2 complexes that were shown in Figure 2—figure supplement 1B of the original submission, as there was no difference between the activity of Mot1 mutants on this substrate in the first place. New data comprise the results of the assays performed on TBP:DNA complexes and, as in the previous version of the assay, show reduced activity of the Mot1^ΔC^ mutant.

5) The reviewers don't think the authors can state "Our model of the Mot1:TBP:NC2:DNA complex illustrates how the Swi2/Snf2 domain of Mot1 engages with upstream DNA," since there is no DNA in the structure. It would be more appropriate to say "…suggests where…" instead of "illustrates how".

We concur with the comment and toned down this statement to “Our model of the Mot1:TBP:NC2:DNA complex suggests where the Swi2/Snf2 domain of Mot1 might engage with upstream DNA”.

[Editors' note: further revisions were requested prior to acceptance, as described below.]

The majority of the reviewer comments have been satisfactorily addressed. Before making a final decision we require one clarification regarding the gel in the new Figure 2D. It appears that in this gel, the ATP-dependent effects seen in lanes 5, 9, 11 and 13 mainly disrupt the ternary complex (M:T:D) and not the TBP-DNA complex (T:D). This seems to be different than in Wollman et al., 2011 and in the previous submission of this work where the T:D complex is disrupted in the presence of ATP and WT Mot1. Please send us a brief paragraph clarifying the reasons for these apparent differences.

Thank you very much for you very positive feedback on our revised manuscript and your invitation to clarify a remaining point regarding the ATP dependent dissociation of TBP:DNA complexes by Mot1. We are very happy to provide a brief explanation for the observed differences. As noted, in the gel shown in Figure 2D (revision 1), the addition of ATP (lanes 5, 9, 11, and 13) mainly resulted in dissociation of the ternary complex (M:T:D) and not the TBP:DNA complex (T:D). In the earlier submission as well as in our previous work (Wollmann et al., 2011), T:D complexes were disrupted in the presence of ATP. We are confident that the differences observed in these different experiments are due to small but significant alterations in the specific experimental conditions. In the experiment shown in Figure 2D (revision 1), conditions were chosen that allowed the T:D complex to still persists so that we could better quantify differences in activity between WT *Ct*Mot1 and the mutants, rather than driving the system to an endpoint. Of note, TBP was added at a concentration of 15 nM and ATP was added for 6 minutes before loading the reactions on the gel. A relatively modest decrease in TBP concentration from 15 nM to 10 nM in combination with an extension of the incubation time with ATP to 10 minutes resulted in disruption of T:D as shown in the gel image below and is consistent with our previously published results. The other components were present at the same concentrations in both experiments: DNA (40 nM), *Ct*Mot1 (25 nM) and competitor DNA (800 nM). We did not use the more efficient dissociation conditions in the experimental results shown in the paper because under the conditions in which all of the labelled DNA is in a free state, it is not possible to see the differences in dissociation catalyzed by WT versus L1658A/Y1659A, R4D, or D1720R. The Mot1, TBP, DNA system is very sensitive to experimental conditions, since multiple equilibria and complexes are in competition (M:T:D, M:T*, T:D, T:T*, * have blocked TBP DNA binding site). To address the remaining concern, we added the new data as new Figure 2D (revision 2), and moved Figure 2D (revision 1) to Figure 2—figure supplement 1B (along with its quantification).